

# Evaluation of change in trabecular bone structure surrounding dental implants by fractal dimension analysis and comparison with radiomorphometric indicators: a retrospective study

Ilkim Karadag[1] and Hasan Guney Yilmaz[2]

[1] Faculty of Dentistry Department of Periodontology, Ankara University, Ankara, Turkey
[2] Faculty of Dentistry Department of Periodontology, Near East University (Cyprus), Mersin, Turkey

## ABSTRACT

**Background:** The trabecular bone surrounding dental implant show some structural changes during healing period. The purpose of this study is to observe the change in trabecular bone with fractal dimensional analysis from baseline to 3rd month of implant placement. It was also aimed to determine the correlation of fractal dimension (FD) change with morphometric indices, mandibular cortical index (MCI) and mandibular cortical width (MCW).

**Methods:** Digital panoramic radiographs taken from 14 patients were evaluated in this study. A total of 30 implants which were placed on mandibular premolar or molar region were included. MCI and MCW assessments were made on baseline radiograph and FD were analyzed on baseline and 3rd month radiograph. FD change by time was recorded separately for every ROI. A paired sample t-test was used to evaluate the change between FD-baseline and FD-3rdmonth values. One-way ANOVA was used to determine the relationship between FD change and MCI. FD change and MCW measurements correlation was calculated by Pearson correlation analysis

**Results:** FD value increased in 75 of 90 implant-adjacent regions evaluated in the 3rd month and it was observed that the FD values were statistically significantly increased in the mesial, distal and apical regions at the 3rd month measurements. There was no statistically significant relationship between mean FD change and MCI, and there was no correlation between mean FD change and MCW value.

**Discussion:** There are many studies evaluating the resorptive changes in bone. However, there are few studies investigating whether there is a correlation between pre- and post-operative trabecular pattern with MCI and MCW. The results of this study indicate that the implants have an improving effect on bone trabeculation in the region where they are placed.

Corresponding author
Ilkim Karadag,
karadagilkim@gmail.com

## INTRODUCTION

All treatments involving dental implants are based on the concept that the implants are attached directly to the surrounding bone tissue and appropriately transmit the incoming occlusal forces to the alveolar bone. The trabecular bone surrounding the implant plays an important role in dispersing the stress caused by chewing forces and supporting the functional pressure exerted by the implant, as it allows load transfer (*Matsunaga et al., 2010*). In other words, the transmission of this stress to the bone is critical for successful dental implant treatment (*Van Staden, Guan & Loo, 2006*). Bone tissue resorption and formation is in a continuous cycle. This cycle causes continuous modeling and remodeling in bone tissue (*Stanford & Brand, 1999*). Thanks to this continuity, it ensures the stability of the bone-implant interface after the placement and loading of the implant. For this reason, it has become important to monitor the bone surrounding the implant during the planning and maintenance stages of implant procedures.

The term fractal is derived from the Latin word *fractus*, meaning fracture. The concept of fractal analysis was introduced to describe similar shapes, curves, surfaces, and repetitive shapes, and was used to describe and measure morphologies in the natural world (*Mandelbrot, 1983*). Fractal analysis is used by medical radiologists to evaluate images independently of parameters such as projection geometry, alignment, and radiodensity (*Buckland-Wright et al., 1994*; *Lynch, Hawkes & Buckland-Wright, 1991*; *Buckland-Wright, Lynch & Bird, 1996*). Fractal dimension (FD) measurements, calculated through fractal analysis, were shown to be relatively insensitive to variations in radiographic angulation, radiodensity, or radiographic machine settings, supporting its use as a diagnostic tool of nonstandardized radiographs (*Jolley, Majumdar & Kapila, 2006*).

Devices used today for measuring bone density and bone mass are expensive and inefficient for routine use in general dental practice (*Kribbs, 1992*; *von Wowern, 1985*). Aluminum step-wedges with periapical radiography is an inexpensive method where we can observe the change in bone density specific to the patient, but this technique is not suitable for obtaining numerical data. *Klemetti & Kolmakow (1997)* introduced the mandibular cortical index in 1997, which would enable the evaluation of the mineral structure of the bone without the need for expensive methods, with the idea of evaluating using readily available equipment such as panoramic radiography. Another radiomorphometric indicator that give results parallel to bone mineral density measurement is the mandibular cortical width (MCW) defined by *Ledgerton et al. (1997)*.

This retrospective study aimed to evaluate the effects of dental implants on the trabecular structure of the alveolar bone around the region where they are placed. It is aimed to reveal the amount of change numerically by performing fractal dimension analysis on radiographies before surgery and 3 months after surgery. In addition, the relationship between mandibular cortical index or mandibular cortical width (Mental Index) measurements, which are important indicators of bone mineralization, and the change in fractal dimension value will be shown.

## MATERIALS & METHODS

The study was designed as a retrospective study and was approved by Ankara University Faculty of Dentistry Ethical Committee (36290600/62). Written consent was not needed to be obtained because of the retrospective design of the study. Panoramic radiographs of 30 implants placed in 14 patients (nine females with 21 implants and five males with nine implants) who applied to Ankara University Faculty of Dentistry, Department of Periodontology with the complaint of missing teeth were evaluated (age ranged from 33 to 75 years). All of these patients had partial edentulous and had not used dentures before. None of the patients included in the study had a systemic disease that would affect bone metabolism or wound healing, and none of them were smokers or chronic alcohol consumers. All implants were placed by the same surgeon according to manufacturer's instruction and without any hard tissue grafting procedure. All of the evaluated implants were bone level implants manufactured by the same company (Dentis OneQ, Daegu, Korea). All implants were in diameters of 3.7 or 4.2 mm and length of 8, 10 or 12 mm.

The exclusion criteria were set as follows:

- Implants placed in the maxilla or anterior to the mental foramen in the mandible.
- Patients with a systemic disease that has an effect on bone tissue or using drugs known to affect bone metabolism.
- Immediate placed or immediate loaded implants.
- Implants that have undergone additional augmentation procedures at the bone crest during placement.
- Cases where the neck of the implant is not completely within the bone crest during placement and tissue level implants.
- Implants that include structures that change the trabecular image, such as teeth, other implants, artifacts, in the apical, mesial or distal ROI areas.
- Cases with incorrect positioning on baseline or control radiographs or in which the image is of unevaluable quality.
- Implants with resorption of the bone adjacent to the implant neck observed on the control radiograph.

Radiographs were obtained with same device (Planmeca Promax, Helsinki, Finland) (54–84 kV, 1–16 mA, 13.8 s,) and the same protocol. The patients were positioned according to the manufacturer's recommendations: the Frankfurt horizontal plane was parallel to the floor, and the sagittal plane was parallel to the vertical line produced by the panoramic device. The images were exported as TIFF files with size of 2,976 × 1,536 pixels. All analyses were conducted in a dim lighted room with a quiet environment.

Mandibular cortical index (MCI) and mandibular cortical width (MCW) measurements were performed on baseline radiographs (Fig. 1). MCI was assessed by observing the mandibular cortical bone from the mental foramen region to the third molar region according to the following classification as suggested by *Klemetti & Kolmakow (1997)*.

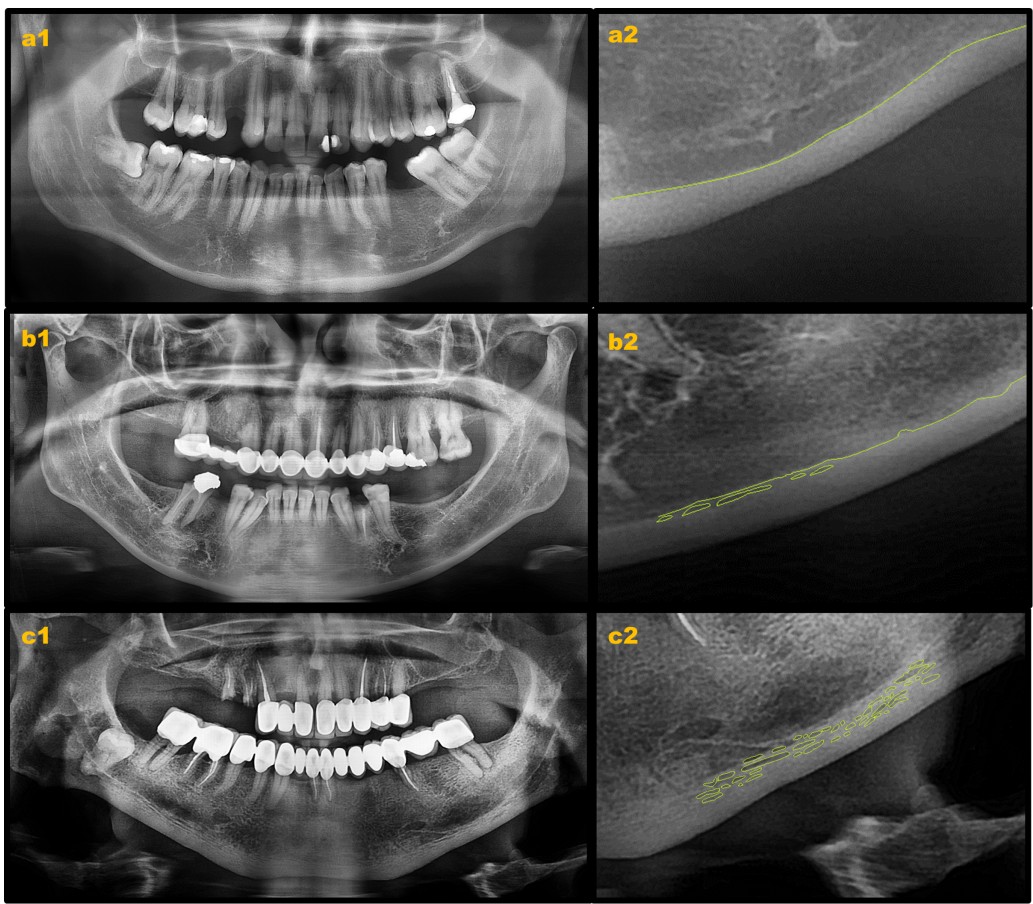

**Figure 1 Mandibular cortical index evaluation on digital panoramic radiographs.** (A) Klemetti C1; (B) Klemetti C2; (C) Klemetti C3.

1. C1: The endosteal margin of the cortex is even, regular, and sharp on both sides of the mandible.
2. C2: The endosteal margin appears to have semilunar defects or has resorptive cavities with cortical endosteal residues one to three layers thick on one or both sides.
3. C3: The endosteal margin consists of numerous (>3) thick cortical endosteal residues and is clearly porous.

MCW refers to the thickness of the cortical layer at the lower edge of the mandible at the level of the mental foramen. For this, the width of the inferior cortical layer will be measured on the line passing through the center of the mental foramen and extending perpendicular to the lower edge of the mandible on panoramic radiographs (*Ledgerton et al., 1997*). MCW in the mental foramen region was measured on both sides of the cortical bone and the mean value of the right and left MCW values was defined for each individual according to methods reported in previous studies (*Ledgerton et al., 1997*; *Benson, Prihoda & Glass, 1991*) (Fig. 2).

All evaluations and measurements were performed by a single observer, using ImageJ software (National Institutes of Health, Bethesda, MD, USA). FD evaluations were made in

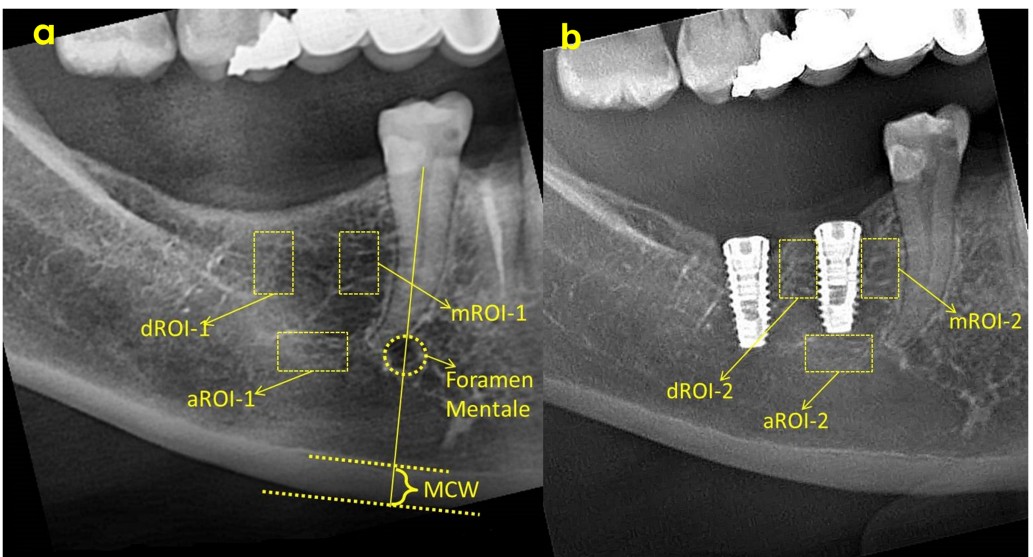

**Figure 2 ROI selection and MCW measurement on baseline radiograph (A) and ROI selection on 3rd month radiograph (B).** ROI, Region of interest; aROI, apical ROI; mROI, mesial ROI; dROI, distal ROI; MCW, mandibular cortical width.

three different areas, mesial, distal and apical of each implant, and fractal dimension analysis was performed by determining the same areas on the pre-surgical radiograph (Fig. 2). As a result, a total of six numerical fractal dimension values (FD) of each implant was obtained. In addition, FD value difference between two images of same implants was calculated and the amount of change was recorded.

Region of interest (ROI) was determined as a vertical rectangle measuring 30 × 60 pixels (h = 60 pixels) in the mesial and distal (mROI and dROI) and a horizontal rectangle of 60 × 30 pixels in the apical. When determining mROI and dROI, it was drawn as a rectangle adjacent to the corners of the threads starting from the most coronal groove level of the implant. The same regions had also been detected in the pre-procedural radiograph of the same implant region with use of replicating feature of software. The point of attention in the selection of ROIs was the positioning of the ROIs in the regions that will completely contain the trabecular structure. In other words, ROIs did not contain implants, roots, resorbed areas, or radiopaque areas. These ROIs was separated from the main image and saved as a separate image file. Using the features of ImageJ software, image processing steps were applied according to the box counting algorithm suggested by White and Rudolph (Fig. 3) (*White & Rudolph, 1999*). These steps were cropping and duplicating the image (Fig. 3A), blurring with Gaussian filter (kernel size 30) (Fig. 3B), subtracting heavily blurred image from original and adding 128 (Fig. 3C), binarization thresholding on a brightness value of 128 (Fig. 3D), eroding (Fig. 3E), dilating (Fig. 3F), inverting (Fig. 3G) and skeletonizing (Fig. 3H). Numerical fractal dimension value (FD) was obtained for each image (Fig. 3I).

After the control session performed in the third month, gingival formers were placed with minor surgery. In the following process, prosthetic restorations were completed using personalized abutments.

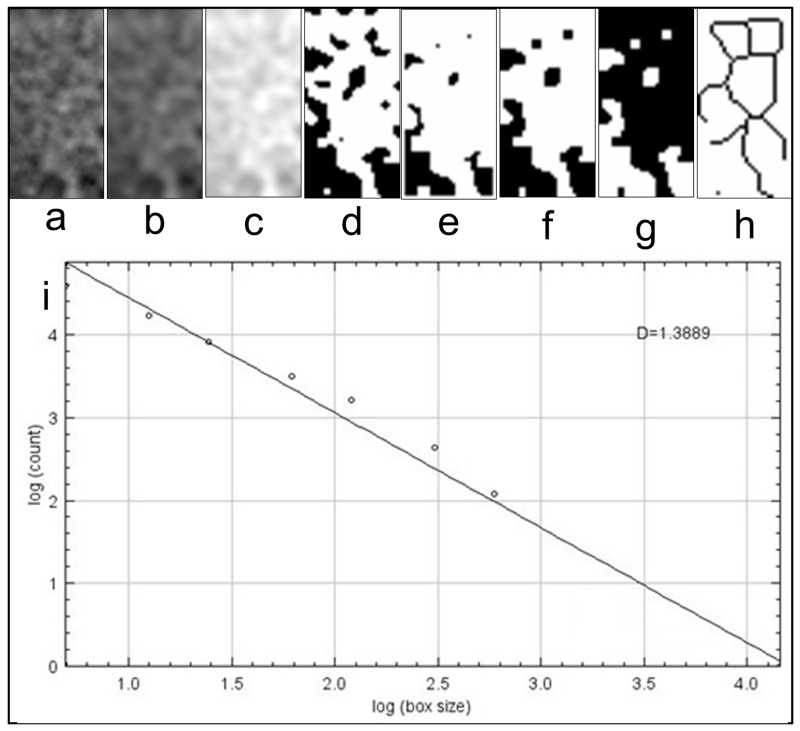

**Figure 3 Fractal dimensional analysis.** (A) Duplicated image from original radiograph; (B) application of Gaussian Blur; (C) Added 128 value; (D) Making image binary; (E) Eroded image; (F) Dilated image; (G) Inverted image; (H) Skeletonized image; (I) Results table.

Prior to study observer was calibrated by re-evaluating 20% of implants. All data obtained from panoramic radiographs was loaded and statistical analyzes were conducted with SPSS software (SPSS for Windows v.24.0, IBM, Armonk, NY, USA). Mean and standard deviation values for FD at baseline (FD-baseline) and 3rd month of surgery (FD-3rdmonth) of 30 implants were determined by descriptive statistics. The normal distribution of data was determined by Shapiro–Wilk test. Homogeneity of variances was evaluated with Levene test. Paired sample t-test was used to evaluate the significance of the change between FD-baseline and FD-3rdmonth values. One-way ANOVA was used to determine the relationship between FD change and MCI, *post hoc* multiple comparisons were evaluated with Bonferroni correction. FD change and MCW measurements correlation was calculated by Pearson correlation analysis. A probability level of less than 5% ($p < 0.05$) was accepted as statistically significant. Power analyses were conducted for the test results that did not reject null hypothesis.

## RESULTS

A total of 180 ROIs were determined for 30 implants, MCI and MCW measurements were made, mean FD of implants and FD changes were calculated for each evaluated region and recorded separately. Of the 14 patients with 30 implants included in the study, nine were female (21 implants) and five were male (nine implants).

**Table 1 Results of Kappa test in intra-observer reliability.**

|  | mROI-1 | dROI-1 | aROI-1 | mROI-2 | dROI-2 | aROI-2 | MCI | MCW |
|---|---|---|---|---|---|---|---|---|
| Kappa values | 0.886 | 0.894 | 0.860 | 0.892 | 0.876 | 0.880 | 0.848 | 0.922 |

**Table 2 Pre- and post-surgical mean fractal dimension (FD) values and mean FD change values and significancy of difference tested with paired sample t-test.**

|  | FD-baseline | | FD-3rdmonth | | FD Change | | *Sig.* |
|---|---|---|---|---|---|---|---|
|  | Mean | St Dev | Mean | St Dev | Mean | St Dev | *p* |
| **Mesial** | 1.333 | 0.105 | 1.386 | 0.082 | 0.053 | 0.07 | <0.001* |
| **Distal** | 1.324 | 0.092 | 1.409 | 0.073 | 0.086 | 0.102 | <0.001* |
| **Apical** | 1.329 | 0.087 | 1.388 | 0.881 | 0.06 | 0.074 | <0.001* |

Note:
  * Significant at $p < 0.005$.

**Table 3 Fractal dimension (FD) change relationship with mandibular cortical index (MCI) score (one-way ANOVA) and mandibular cortical width (MCW) (Pearson correlation analysis).**

|  | MCI | MCW | |
|---|---|---|---|
|  | *P* | R | *p* |
| **Mesial** | 0.183* | 0.203 | 0.140* |
| **Distal** | 0.298* | 0.303 | 0.228* |
| **Apical** | 0.208* | 0.230 | 0.159* |

Notes:
  * Not significant at $p < 0.05$.
  R, Pearson correlation coefficient.

For repeated measurements and evaluations Kappa coefficient were indicating good intra-observer reliability (Table 1).

When MCI was evaluated, 12 out of 30 (40%) implants were found to be placed on C1 bone, 12 (40%) were on C2, and 6 (20%) were on C3 bone. Mean MCW value was 3.43 ± 0.29 (ranged from 2.91 to 3.86).

While FD value increased in 75 of 90 implant-adjacent regions evaluated in the 3rd month, the FD value remained the same in 1 region and the FD value increased in 14 region. Mean FD values before implant placements were 1.333 at mesial, 1.324 at distal and 1.329 at apical and increased to 1.386 at mesial, 1.409 at distal and 1.388 at apical after 3 months of implant placement (Table 2).

As a result of the comparison of the FD-baseline–FD-3rdmonth differences with the paired sample t-test, it was observed that the FD values were statistically significantly increased in the mesial, distal and apical regions at the 3rd month measurements.

It was observed that there was no statistically significant relationship between FD change and MCI, and there was no correlation between FD change and MCW value (Table 3).

## DISCUSSION

The concept of osseointegration is defined as a successful and functional direct connection between the bone and the implant surface (*Brånemark, 1983*). The success of the osseointegration process depends on factors such as the biocompatibility of the alloy used in production, the macro and microstructure of the implant, the surgical technique performed and the bone quality. The primary stability of the implant during insertion has been considered a prerequisite for its survival (*Elias, Brunski & Scarton, 1996*). In addition, being able to evaluate the implants whose prosthetic loading time has come can give an important idea about the next stages of the treatment. For this purpose, one of the most frequently used methods for the evaluation of the trabecular structure adjacent to the implant is resonance frequency analysis, but this method requires additional equipment (*Simunek et al., 2010*). Evaluation of trabecular structure with fractal analysis is a subject that has been evaluated many times in recent years and its ease of application is an important advantage (*Kış & Güleryüz Gürbulak, 2020*; *Kulczyk, Czajka-Jakubowska & Przystańska, 2018*). In a study, fractal analysis values were compared with insertion torque and resonance frequency analysis values, as a result, it was shown that FD gave correlative results with both evaluation methods (*Suer, Yaman & Buyuksarac, 2016*).

Fractal analysis has different uses such as periodontitis, bruxism and evaluating changes in bone structure caused by orthodontic treatment (*Updike & Nowzari, 2008*; *Eninanç, Yalçın Yeler & Çınar, 2021*; *Cesur et al., 2020*). In addition to panoramic radiographs, fractal analysis can be successfully evaluated on periapical radiographs and cone beam computed tomography images (CBCT) (*Jolley, Majumdar & Kapila, 2006*; *Hua et al., 2009*). However, CBCT images have low resolution and the result of fractal analysis may be affected by the resolution of the images. Therefore, the use of panoramic radiographs for fractal analysis is recommended (*Magat & Ozcan Sener, 2019*).

*Sansare, Singh & Karjodkar (2012)* showed that the FD value increased after implant placement due to the increase in the surrounding bone microstructure and the amount of bone trabeculae. The ROI was placed near the apical end of the implant and a square of $80 \times 80$ pixels was selected as an ROI in the mentioned study, but in our study three ROIs were selected for each implant with different ROI size and shape. We chose to use a narrower ROI in a rectangular structure to avoid inclusion of adjacent anatomical structures on all implants. Previous studies have suggested that the ROI location is more critical than its size for fractal analysis procedure (*Majumdar, Weinstein & Prasad, 1993*; *Shrout, Potter & Hildebolt, 1997*).

*Heo et al. (2002)* also reported that FD decreased on the second day after orthognathic surgery and FD gradually increased over time. *Soylu et al. (2021)* evaluated trabeculation around implants using fractal analysis on panoramic radiographs obtained in short-period intervals and concluded FD was significantly lower in first week measurements. This can be explained by the healing pattern of the bone. *Mu et al. (2013)* showed that FD values increased significantly in the evaluations made 12 months after the implant surgery. *Wilding (1995)* screened FD values after a longer recall period. They evaluated

panoramic radiographs of 18 patients and a significant increase in fractal dimension was found during the period up to 2 years after surgery. The most pronounced increase was in the region of bone around the neck of the implant. Therefore we placed rectangle shaped ROIs to the most coronal side of implants for obtaining mesial and distal FD values.

Branemark stated that after implant placement, a recovery period of at least 3 to 6 months is required for prosthetic loading (*Branemark, Zarb & Alberkstsson, 1985*). In some studies, the FD change in the peri-implant alveolar bone was evaluated by including the values obtained after the prosthetic loading of the implants with multiple measurements (*Wilding, 1995*; *Zeytinoğlu, 2015*). In these studies, it was reported that FD showed a significant increase over time after surgery. In our study, we evaluated radiographs taken at the beginning and at the third month after surgery. The reason for this is to evaluate the idea that "implants placed with standard procedure are ready for prosthetic loading at the end of the third month" by observing the change that can be seen in the fractal value.

It has already been recognized that all fractal analysis studies are full of parameters that are difficult to control (*Shrout, 1998*). It has also been recognized that different methods used to estimate FD may not agree in their results, and this remains a gray area for which there is currently no universally acceptable answer (*Caligiuri, Giger & Favus, 1994*). Additionally it has been suggested that differences in experimental design may cause FD to differ (*Lee et al., 2010*). We used box-counting method of the ImageJ software for fractal dimension analysis in our study because it was readily available and easy to use.

MCW is the only quantitative index with high intraobserver and interobserver agreement and moderate correlation. It is considered as one of the most accurate indicators to predict women with osteoporosis and in need of treatment (*Leite et al., 2010*). While MCW may not reflect increased bone turnover after menopause, it can be used more successfully when reflecting peak bone mass at a young age. *Devlin & Horner (2002)* reported that thinning of the mandibular cortex is associated with low skeletal bone mineral density in a normal perimenopausal woman. In our study, a value below 3 mm was not found in the MCW measurement of any patient.

In the literature, there are many studies evaluating changes in trabeculation as a result of early loading, bone quality, peri-implantitis areas and primary stability of the implant using methods such as fractal dimensional analysis, MCI evaluation and resonance frequency analysis (*Hayek et al., 2020*; *Lee et al., 2010*; *Friberg et al., 1999*). However, there are few studies investigating whether there is a correlation between pre- and post-operative trabecular pattern with MCI and MCW (*Bayrak et al., 2020*; *Ersu, Akyol & Etöz, 2021*). In one of these studies *Ersu, Akyol & Etöz (2021)* found no correlation between FD and MCI but there were significant differences in FD among MCW measurements. *Kato et al. (2020)* evaluated FD, MCW, and MCI on panoramic radiographs of patients with cemento-osseous dysplasia (COD) (*Kato et al., 2020*). The study, in which different ROI shapes and locations were also compared, showed that the mean of MCW and FD in the COD group was significantly lower, and in the MCI assessment, significantly more people were included in the C3 class at the COD group. Similarly, *Bayrak et al. (2020)*

reported that MCW and FD were significantly lower in thalassemia patient. In our study, we questioned the existence of a relationship between the increase in FD observed around the implant and MCI and MCW.

MCI and MCW measurements on currently available panoramic radiographs have been shown to be useful tools to gain an idea of bone density. However, these methods have some limitations. Radiographs with different image quality and magnifications may affect validity (*Calciolari et al., 2015*). The most important limitation is the consistency of the measurements depends on the experience of the observer. Experienced specialist physicians have been reported to have significantly higher inter-observer consistency than final year dental students (*Jowitt et al., 1999*). In addition, it was determined that MCI evaluation performed on panoramic radiographs of patients with osteopenia or osteoporosis was more consistent than those performed on radiographs of control group patients (*Halling et al., 2005*).

There are some limitations of this study; total number of implants and patients included were low because of the retrospective design of the study. Additionally as including the implants with different dimensions, FD change of different surfaces or different sizes of implants could not be compared. Conducting controlled clinical studies on FD change with a larger sample size and certain implant dimension would provide more reliable information about the structural changes of the bone in order to examine the effects of diseases affecting the bone cavity such as osteoporosis. Another point that should be emphasized is that the FD and MCI or MCW correlations in the study were not considered as a priority in the evaluation. In order to more accurately determine the relationship between these values and the FD change, studies can be conducted in which the sample groups are classified according to the MCI index or MCW measurements. Conducting a similar study on digital 3D models obtained by CBCT imaging would have allowed the 3-dimensional evaluation of the change in bone tissue.

When power analysis was performed to calculate the sample size during the study design phase, we saw that a sample size of 29 implants would be sufficient for 95% power. We saw that the power of the study was calculated as 0.9999627 with the *post hoc* power analysis we performed using the data we obtained in study. However, when the distribution of men and women is examined, it is seen that there is no equality. This can be seen as a challenge due to the retrospective design of our study. It may be possible to investigate the relationship between FD change and gender by designing a study involving equal numbers of male and female subjects.

## CONCLUSIONS

This study provides evidence that the peri-implant trabecular structure develops relative to the baseline by the third month, which is the decision-making stage for the insertion of healing caps, the second surgical stage in implant procedures. It has also been shown that there is no significant correlation between FD change and MCI or MCW values, which are important morphometric indicators for bone density. Designing studies in which the peri-implant trabecular structure can be observed in shorter time intervals will contribute

to a more accurate understanding of the timing of bone resorption and apposition events following the initial placement of the implant.

### Funding
The authors received no funding for this work.

### Competing Interests
The authors declare that they have no competing interests.

### Author Contributions
- Ilkim Karadag conceived and designed the experiments, performed the experiments, analyzed the data, prepared figures and/or tables, authored or reviewed drafts of the paper, and approved the final draft.
- Hasan Guney Yilmaz conceived and designed the experiments, authored or reviewed drafts of the paper, and approved the final draft.

### Human Ethics
The following information was supplied relating to ethical approvals (*i.e.*, approving body and any reference numbers):

The Ankara University Faculty of Dentistry Ethical Comitee approved the study (36290600/62).

### Data Availability
The raw measurements are available in the Supplemental File.

### Supplemental Information
Supplemental information for this article can be found online at http://dx.doi.org/10.7717/peerj.13145#supplemental-information.

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
