# Peer review of "Evaluation of change in trabecular bone structure surrounding dental implants by fractal dimension analysis and comparison with radiomorphometric indicators: a retrospective study"

_PeerJ, doi:10.7717/peerj.13145_

## Round 0.1 · original submission · Major Revisions

Thank you to the authors for their submission. Please find the attached comments from the reviewers and answer them accordingly. I look forward to receiving your revision.

Reviewer 1 ·

Basic reporting

Comments are attached.

Experimental design

Comments are attached

Validity of the findings

Comments are attached

Additional comments

Comments are attached

Annotated reviews are not available for download in order to protect the identity of reviewers who chose to remain anonymous.

Reviewer 2 ·

Basic reporting

In this study, Karadag et al. aim to investigate the correlation of fractal dimension change with morphometric indices, mandibular cortical index, and mandibular cortical width and their correlation pre-and post-operatively in a retrospective study. Based on the collected data, they found, as expected, FD values were increased in the mesial, distal, and apical regions at the 3rd-month measurements. However, no relationship between FD and MCI or MCW was observed. Although the experimental design is easy to follow and the results are scientifically fair, the study failed to show a robust, innovative approach.

Experimental design

Although many studies in the literature have extensively investigated the same area, the presented methodology is acceptable for answering the research question.

Suggestions:

The author listed that the systemic conditions were challenging to have ( whereas it is clinically not acceptable to miss such data), the status, e.g., fully dentate or partially dentate edentulous, is critical to include.

Importantly, the Manufacturers of included implants are critical to list.

Authors are encouraged to use a Full PANORAMIC X-Ray in addition to the magnified ones in FIG. 1

Validity of the findings

Findings and conclusion are fair and clearly linked to the original question.

Additional comments

1. Proofreading for the English language is suggested.

2. The limitation of the study should specify those panoramic radiographs represent a two-dimensional image of a three-dimensional structure. Although CBCT is not routinely used in every patient who presents to a dentist, the volumetric analysis provides valuable data. Three-dimensional analysis performed with CBCT will undoubtedly give more valuable volumetric data.

Reviewer 3 ·

Basic reporting

The text is written in correct English.
The selection of literature items is appropriate, and these are relatively new items.
Figure 1 regarding the Klemetti method is probably redundant unless the authors indicate or draw on the figure which criteria they used to assign to each category
Table 1 and Figure 2 should be adjusted to be more understandable.

Experimental design

The authors used the Klemetti method, which was originally used to evaluate 77 postmenopausal women. Therefore, they should indicate this fact as a limitation of this method. Alternatively, authors can refer to other studies in which the age and gender factors have been verified.
The structure of the study group, demographic data and gender should be placed in the material and methods section.
The description of the test group is superficial. Factors such as general health, comorbidities and habits, including smoking, were not considered.
In the beginning, the authors distinguish 3 ROIs: mesial, distal and apical, for which they define the fractal dimension. However, for comparative studies, they calculate the arithmetic mean from all three regions. It isn't easy to understand the basics of doing this. While the mesial and distal regions have similar characteristics and their connection makes sense, the apical region is different.
The criteria for the location of the ROIs on both sides of the implant are unclear. What if the area includes compacted and spongy bone? What if the bone level around the implant has lowered after a healing period of 3 months?

Validity of the findings

Table 1 is hardly readable, the position of column “total” in the table is illogically located.
Figure 2 does not explain the doubts about the precise location of the areas, aROI-2 contains a small fragment of the implant in its region - this is probably an unintended situation and should be adjusted correctly
The authors report that the study included implants of various sizes. Was this factor statistically insignificant in the evaluation of the results?

Additional comments

In the abstract section, I suggest that you omit the part about the discussion. Moreover, the last sentence of the discussion is a re-description of the result described in the results section.
Authors should speculate more about why MCI and MCW parameters correlate with the FD dimension change. Is the attempt to link the mean FD dimension with MCI and MCW not the weak point of this study?

---

## Round 0.2 · accepted · Accept

Thank you for the detailed revision. The quality of the manuscript has improved significantly and has been accepted for publication.

Reviewer 3 ·

Basic reporting

Errors in the English language have been corrected
The selection of literature items is appropriate, and these are relatively new items.
Figure 1 regarding the Klemetti method was improved with the necessary details placed within the x-ray images.
Tables 1 and 2 were adjusted to be easier to understand and follow the manuscript's main text.

Experimental design

The authors used the Klemetti method and successfully clarified the intention to do so despite the different age/sex characteristics of the study group.
The structure of the study group, demographic data and gender were moved in the material and methods section.

Information about the position of ROI and the intention for separate statistical calculations was explained.

Validity of the findings

Table 1 and Figure 2 were corrected according to suggestions

Additional comments

Suggestions from the initial review are now included in the manuscript; I also welcome the changes suggested by other reviewers